# Scalable Online Planning
# via Reinforcement Learning Fine-Tuning

**Arnaud Fickinger**[*]
Facebook AI Research
arnaudfickinger@fb.com

**Hengyuan Hu**[*]
Facebook AI Research
hengyuan@fb.com

**Brandon Amos**
Facebook AI Research
bda@fb.com

**Stuart Russell**
UC Berkeley
russell@berkeley.edu

**Noam Brown**
Facebook AI Research
noambrown@fb.com

## Abstract

Lookahead search has been a critical component of recent AI successes, such as in the games of chess, go, and poker. However, the search methods used in these games, and in many other settings, are tabular. Tabular search methods do not scale well with the size of the search space, and this problem is exacerbated by stochasticity and partial observability. In this work we replace tabular search with online model-based fine-tuning of a policy neural network via reinforcement learning, and show that this approach outperforms state-of-the-art search algorithms in benchmark settings. In particular, we use our search algorithm to achieve a new state-of-the-art result in self-play Hanabi, and show the generality of our algorithm by also showing that it outperforms tabular search in the Atari game Ms. Pacman.

## 1 Introduction

Lookahead search has been a key component of successful AI systems in sequential decision-making problems. For example, in order to achieve superhuman performance in go, chess and shogi, AlphaZero leveraged Monte Carlo tree search (MCTS) [38]. MuZero extended this even further to Atari games, again using MCTS [32]. Without MCTS, AlphaZero performs below a top human level, and more generally no superhuman Go bot has yet been developed that does not use some form of MCTS. Similarly, search algorithms were a critical component of AI successes in backgammon [45], chess [10], poker [27, 7, 8], and Hanabi [22]. However, even though different search algorithms were used in each domain, *all* of them were *tabular* search algorithms, i.e., a distinct policy was computed for each state encountered during search, without any function approximation to generalize between similar states.

While tabular search has achieved great success, particularly in perfect-information deterministic environments, its applicability is clearly limited. For example, in the popular partially observable stochastic AI benchmark game Hanabi [5], one-step lookahead search involves a search over about 500 possible next states. However, searching over all two-step joint policies would require a search over $20^{20}$ states, which is clearly intractable for tabular search. Additionally, unlike perfect-information deterministic games where it is only necessary to search over a tiny fraction of the next several moves, partial observability and stochasticity make it impossible to limit the search to a tiny subset of all possible states. Fortunately, many of these states are extremely similar, so a search algorithm can in theory benefit by generalizing between similar states. This is the motivation for our non-tabular search algorithm.

---

[*]Equal Contribution.

35th Conference on Neural Information Processing Systems (NeurIPS 2021).

In this paper we take inspiration from related research in continuous control environments that use non-tabular planning algorithms to improve performance at inference time [47, 25, 1]. These methods leverage finite-horizon model-based rollouts. Specifically, we replace tabular search with fine-tuning of the policy network at inference time. We show that with this approach we are able to achieve state-of-the-art performance in Hanabi. Specifically, our method is able to search multiple moves ahead and discover joint deviations, which in general is intractable using tabular search. We also show the generality of our approach by showing that it outperforms Monte Carlo tree search in deterministic and stochastic versions of the Atari game Ms. Pacman.

## 2 Background

### 2.1 MDPs, POMDPs, and Dec-POMDPs

We consider Markov decision processes (MDPs) [42], where the agent observes the state $s_t \in S$ at time $t$, performs an action according to their policy $a_t \sim \pi(s_t)$ and receive reward $r_t = r(s_t, a_t)$. The environment transit to the next state following transition probability $s_{t+1} \sim P(s_{t+1}|s_t, a_t)$.

POMDPs [39] extend MDPs to the partially observable setting where the agent cannot observe the true underlying state but instead receive information through a observation function $o_t = \Omega(s_t)$. Due to the partial observability, the agent policy $\pi$ often needs to take into account the entire action-observation history (AOH) denoted as $\tau'_t = \{o_0, a_0, r_0, \ldots o_t\}$ with $\tau_t = \{s_0, a_0, r_0, \ldots, s_t\}$ representing the true underlying trajectory.

Dec-POMDPs [29, 5] extend POMDPs to the cooperative multi-agent setting. At each time step $t$, each agent $i$ receives their individual observation from the state $o_t^i = \Omega^i(s_t)$ and selects action $a_t^i$. The joint action of $N$ player is a tuple $\mathbf{a_t} = (a_t^1, a_t^1, ..., a_t^N)$ and can be observed by all players. We denote the individual policy for each player as $\pi_i$ and joint policy as $\pi$. We use $\tau_t^i = \{o_0^i, \mathbf{a_0}, r_0, \ldots o_t^i\}$ to represent the AOH from the perspective of agent $i$.

### 2.2 Tabular Search in Dec-POMDPs

SPARTA [22] is a tabular search algorithm previously applied in Dec-POMDPs that is proven to never hurt performance and in practice greatly improves performance. Specifically, SPARTA assumes that all agents in a game agree on a joint blueprint policy $\pi_{\text{bp}}$ and that one or more agents may choose to deviate from $\pi_{\text{bp}}$ at test time. We assume a two player setting in the following discussion. There are two versions of SPARTA.

In **single-agent SPARTA**, only one agent may deviate from the blueprint. We denote the search agent as agent $i$ and the other agent as $-i$. Agent $i$ maintains their private belief over the trajectories they are in given their AOH, $B^i(\tau_t) = P(\tau_t|\tau_t^i)$. At action selection time, agent $i$ compute the Monte Carlo estimation for the value of each action assuming both players will follow blueprint policy $\pi_{\text{bp}}$ thereafter,

$$Q(\tau_t^i, a_t) = E_{\tau_t \sim B^i(\tau_t), \tau_T \sim P(\tau_T|\tau_t, a_t, \pi_{bp})} R_t(\tau_T), \tag{1}$$

where $R_t(\tau_T) = \sum_{t'=t}^{T} r_{t'}$ is the forward looking return from $t$ to termination step $T$. Agent $i$ will pick $\arg\max_{a_t} Q(\tau_t^i, a_t)$ instead of the blueprint action $a_{bp}$ if $\max_{a_t} Q(\tau_t^i, a_t) - Q(\tau_t^i, a_{bp}) > \epsilon$.

In **multi-agent SPARTA**, both agents are allowed to search at test time. However, since agent $i$'s belief over states depends on agent $-i$'s policy, it is essential that agent $i$ replicate the search policy of agent $-i$. Since agent $i$ does not know the private state of agent $-i$, agent $i$ must therefore compute agent $-i$'s policy for *every* possible AOH $\tau_t^i$ agent $i$ may have seen, a process referred to as *range-search*. Once a correct belief is computed, agent $-i$ runs their own search. The range-search can be prohibitively expensive to compute in many cases because players might have millions of potential AOH. For example, Hanabi might have up to about 10 million AOHs. To mitigate this problem, the authors introduce a way to execute multi-agent search only when the range of AOHs is small enough that it is computationally feasible to do so, and the algorithm uses single-agent search otherwise. Even so, SPARTA multi-agent search is limited to *one-ply* search – each agent calculates the expected value of only the next action. The computational complexity grows exponentially with each additional ply. In contrast, our non-tabular multi-agent search technique makes it possible to conduct 2-ply, and deeper, search.

## 2.3 Reinforcement Learning

Our goal is to learn a stationary policy $\pi(a|s)$ such that the expected non-discounted return $J(\pi) = \mathbb{E}_{s_0 \sim \rho(s)} \mathbb{E}_\pi \sum_{t=0}^{\infty} r(s_t, a_t)$ is maximized, where $\rho(s)$ is the initial state distribution. For any policy $\pi$ we define its value $V_\pi(s_t) = \mathbb{E}_\pi(\sum_{l=0}^{\infty} r(s_{t+l}, a_{t+l})|s_t)$, its Q-value $Q_\pi(s_t, a_t) = \mathbb{E}_\pi(\sum_{l=0}^{\infty} r(s_{t+l}, a_{t+l})|s_t, a_t)$ and its advantage $A_\pi(s_t, a_t) = Q_\pi(s_t, a_t) - V_\pi(s_t)$. In reinforcement learning (RL), the transition probability $p$ and the reward $r$ are unknown, and the policy is learned by sampling trajectories $\tau = ((s_0, a_0), ..., (s_T, a_T))$ from the environment.

In Q-learning, we use the sampled trajectories to learn the Q-function of the optimal policy, given by the Bellman equation: $Q(s, a) = \mathbb{E}_{s'}(r(s, a) + \gamma \max_{a'} Q(s', a'))$. The learned policy is greedy with respect to $Q$: $\pi(a|s) = \arg\max_a Q(s, a)$. In DQN, the Q-value is approximated with a neural network $Q_\theta$ that is trained by taking gradients of the mean squared Bellman error:

$$\nabla_\theta \hat{\mathbb{E}}(Q_\theta(s, a) - (r(s, a) + \gamma \max_{a'} Q_{\theta'}(s', a')))^2 \tag{2}$$

where $\theta'$ is a fixed copy of $\theta$ and $\hat{\mathbb{E}}$ denotes the empirical average over a finite number of batches.

In policy gradient (PG), we directly learn a parametrized policy $\pi_\theta$ by performing gradient descent using an estimator of $J$:

$$\nabla_\theta \hat{\mathbb{E}}(\log \pi_\theta(a|s) \hat{A}(s, a)) \tag{3}$$

where $\hat{A}$ is an estimate of the advantage of $\pi_\theta$.

## 2.4 Decision-Time Planning

To make better decisions at test time, we can specialize the learned policy, which we refer to as the **blueprint** policy (also known as the prior policy), to the current state by using it as prior in an online planning algorithm.

One of the most popular planning algorithms is **Monte Carlo tree search (MCTS)**, which has famously enabled superhuman performance in go, chess, and many other settings [36, 37, 38, 32]. MCTS builds a tree of potential future states starting from the current state. Each node $s$ keeps track of its action-state visitation counts $N(s, a)$ and an estimate of its Q-values $Q(s, a)$, refined each time the node is visited. The prior policy is used to select the next action to take during the tree traversal.

MCTS don't scale well in environments with large branching factors caused by a high amount of stochasticity or partial observability. For example, MCTS becomes very expensive as the size of the action space increases because it needs to build an explicit tree to keep statistics about state-action pairs, and nodes need to be visited several times for the method to be effective. Furthermore, MCTS can in the worst case expand very deeply a branch that it misidentifies to be optimal due to a lack of exploration – exacerbated in high dimensions– which leads to a worst-case sample complexity worse than uniform sampling [28].

# 3 Reinforcement Learning Fine-Tuning

To alleviate the inefficiency of tabular search methods like MCTS in large state and action spaces, we propose to formulate online planning as a RL problem that should be quickly solvable given a sufficiently optimal blueprint policy. We do so by using RL as a multiple-step policy improvement operator [12]. More specifically, we bias the initial state distribution towards the current state $s^*$ and we reduce the horizon of the problem, leading to the following truncated objective:

$$\max_{\tilde{\pi}} (\mathbb{E}_{s_0 \sim \rho_{s^*}(s)} \mathbb{E}_{a_{0:H-1}, s_{1:H} \sim \tilde{\pi}} \sum_{t=0}^{H-1} (r(s_t, a_t)) + \mathbb{E}_{a_{H:\infty}, s_{H+1:\infty} \sim \pi} \sum_{t=H}^{\infty} (r(s_t, a_t))) \tag{4}$$

where $\rho_{s^*}(s) = \mathbb{1}(s = s^*)$ and $\pi$ is the blueprint policy. If $\pi$ is optimal, then $\pi' = \pi$ is a solution to the problem. We call this procedure RL Fine-Tuning or RL Search and we use the two terms interchangeably in the remaining.

The blueprint $\pi$ is either directly parameterized by $\theta$ or is greedy with respect to a Q-value parameterized by $\theta$. In both cases, we improve $\pi$ for the next $H$ steps by following the gradient of the

truncated objective with respect to $\theta$. We present the two approaches in more detail in the following subsections.

## 3.1 Policy Fine-Tuning

In policy fine-tuning, we use any actor-critic algorithm to train a blueprint policy and a value network. For our experiments in the Atari environment, we use PPO [34]. At action selection time, to solve objective (4) we use the blueprint policy to initialize the online policy and the blueprint value to truncate our objective:

$$\max_{\tilde{\theta}}(\mathbb{E}_{s_0 \sim \rho_{s^*}(s)}\mathbb{E}_{a_{0:H-1}, s_{1:H} \sim \pi_{\tilde{\theta}}} \sum_{t=0}^{H-1}(r(s_t, a_t)) + V_\phi(s_H)) \tag{5}$$

If $V_\phi$ is a perfect estimator of $V_{\pi_\theta}$, then we exactly optimize the truncated objective 4.

The fine-tuned policy $\pi_{\tilde{\theta}}$ is obtained by performing $N$ gradient steps with the PPO objective. The policy improvement step via policy gradient is presented in Algorithm 1.

---

**Algorithm 1:** Policy Gradient Improvement. We use a standard PPO update to fine-tune the blueprint policy $\pi_\theta$ and value $V_\phi$ from some state $s_t$.

---

**Input** : current state $s_t$, number of updates $N$, global policy parameter $\theta$, global value parameter $\phi$, horizon $H$, number of rollouts $M$

**Output** : updated parameter $\tilde{\theta}$

**Init:**
    $\theta_0 = \theta, \phi_0 = \phi$

**for** $i \leftarrow 1$ **to** $N$ **do**
    Collect M trajectories of H time steps starting from $s_t$ using $\pi_{\theta_{i-1}}$.
    Compute the generalized advantage estimate using:

$$\delta_t = r_t + \gamma V_{\phi_{i-1}}(s_{t+1}) - V_{\phi_{i-1}}(s_t) \forall t \in [0, H-2]$$
$$\delta_{H-1} = r_t + \gamma V_\phi(s_H) - V_{\phi_{i-1}}(s_t) \tag{6}$$

    $(\phi_i, \theta_i) \leftarrow PPO(\phi_{i-1}, \theta_{i-1})$

**return** $\theta_N$

---

## 3.2 Q-value Fine-Tuning

In Q-value fine-tuning, we train a Q-network using any offline RL algorithm. At action selection time, to solve objective (4) we use the blueprint Q-value to truncate our objective:

$$\max_{\tilde{\pi}}(\mathbb{E}_{s_0 \sim \rho_{s^*}(s)}\mathbb{E}_{a_{0:H-1}, s_{1:H} \sim \tilde{\pi}} \sum_{t=0}^{H-1}(r(s_t, a_t)) + \max_a Q_\theta(s_H, a)) \tag{7}$$

The online policy $\tilde{\pi}$ is greedy with respect to the fine-tuned Q-network $Q_{\tilde{\theta}}$ obtained by performing $N$ gradient steps with the mean squared Bellman error. To alleviate instabilities, the transitions used to fine-tune the Q-network are sampled with probability $p$ from the global buffer replay and with probability $1 - p$ from the trajectories sampled at action selection time. The Q-value improvement step is presented in Algorithm 2.

## 3.3 Scaling Multi-Agent Search in Dec-POMDPs with RL Search

### 3.3.1 Scaling Single-Agent RL Search

In a test game, agent $i$ maintains a belief over possible trajectories given their own AOH, $B^i(\tau_t) = P(\tau_t|\tau_t^i)$. At action selection time, agent $i$ performs Q-value fine-tuning (Algorithm 2) of the

---
**Algorithm 2:** Q-Value Improvement. We use a standard Bellman residual update to fine-tune the blueprint Q function $Q_\theta$ from some state $s_t$.

---
**Input** : current state $s_t$, number of updates $N$, global Q-network parameter $\theta$, horizon $H$, number of rollouts $M$, batch size $B$
**Output** : updated parameter $\tilde{\theta}$

**Init:**
   $\theta_0 = \theta$

Collect M trajectories of H time steps starting from $s_t$ using an $\epsilon$-greedy policy wrt $Q_\theta$.
For each trajectory, if the environment is not terminated, replace $r_{t+H-1}$ with
  $r_{t+H-1} + \max_a Q_\theta(s_{t+H}, a)$
**for** $i \leftarrow 1$ **to** $N$ **do**
   Sample B transitions with probability $p$ from the global buffer and probability $1-p$ from the $M$ collected trajectories.
   $\theta_i \leftarrow \nabla_{\theta_{i-1}} \hat{\mathbb{E}}(Q_{\theta_{i-1}}(s, a) - (r(s, a) + \gamma \max_{a'} Q_{\theta'}(s', a')))^2$
**return** $\theta_N$

---

blueprint policy $\pi_{\text{bp}}$ on the current belief $B(\tau_t^i)$. This produces a new policy $\pi^*$. Next, we evaluate both $\pi^*$ and $\pi_{\text{bp}}$ on $E$ trajectories sampled from the belief. If the expected value of $\pi^*$ is at least $\epsilon$ higher than the expected value of $\pi_{\text{bp}}$, the agent plays according to $\pi^*$ for the next $H$ moves and search again at the $(H+1)$th move. Otherwise, the agent sticks to playing according to $\pi_{\text{bp}}$ for the current move and searches again on their next turn.

### 3.3.2 Scaling Multi-Agent RL Search

A critical limitation of single-agent SPARTA, multi-agent SPARTA, and single-agent RL search is that the searching agent assumes the other agent will follow the blueprint policy on all future turns. It is thus impossible for those methods to find *joint* deviations $(a_t^{i*}, a_{t_1}^{-i*})$ where it is beneficial for agent $i$ deviate to action $a_t^{i*}$ if and only if the other agent $-i$ deviates to $a_{t+1}^{-i*}$, an action that will not be selected under blueprint. Searching for joint deviation with general tabular methods is computationally infeasible in large Dec-POMDPs such as Hanabi because the effective branching factor is about $20^{20}$ in Hanabi.

RL search enables searching for the next $H$ moves for all agents jointly conditioning on the common knowledge. The observations in a Dec-POMDP can be factorized into private observation and public observations $o_t^{pub}$. For example, in Hanabi the private observations are the teammates hands and the public observation are the played cards, discarded cards, hints given, and previous actions. The public observation is common knowledge among all players. We can define the common knowledge public belief as $B^{pub}(\tau_t^{pub}) = \{\tau_t | \Omega^{pub}(\tau_t) = \tau_t^{pub}\}$ where $\tau_t^{pub} = \{o_0^{pub}, a_1, r_1, \ldots, o_t^{pub}\}$ and $\Omega^{pub}$ is the public observation function. We can then draw $\tau \sim B^{pub}(\tau_t^{pub})$ repeatedly. The new policy is obtained via Q-value fine-tuning, in the same way as in single-agent RL search. Since the search procedure uses only public information, it can in principle be replicated by every player independently. However, to simplify the engineering challenges of our research, we conduct search just once and share the solution with all the players. If the newly trained policy is better than the blueprint by at least $\epsilon$, every player will use it for their next $H$ moves.

## 4 Experiments

### 4.1 Hanabi Experiments

Hanabi is a 2-5 player partially observable fully cooperative card game and a popular Dec-POMDP benchmark. A detailed description of the rules of the game and an explanation of its challenges can be found in [5]. The deck in Hanabi consists of 5 color suites and each suite contains 10 cards with three 1s, two 2s, two 3s, two 4s and one 5. At the beginning of the game each player draws 4 or 5 cards. The main twist of the game is that players cannot see their own hand but instead observe all other players' hands. All players work together to play cards from 1 to 5 in order for each suite. At each turn, the active player may choose to give a hint to another player at the cost of a hint token.

However, the players start with only 8 available hint token, and must discard a card in order to gain a new hint token.

In recent years, there has been tremendous progress in developing increasingly strong agents for Hanabi [5, 13, 22, 18, 19, 20] to the point where the latest agents are approachingn near-perfect scores. As a result, it is difficult to distinguish performance differences between different techniques. For this reason, we additionally evaluate our technique on a harder version of Hanabi in which the maximum number of hint tokens is 2 rather than 8.

We test both single-agent RL search and multi-agent joint RL search in 2-player Hanabi. Blueprint policies are trained with Q-learning and the same learning method is used for test-time fine-tuning. We compare our methods against SPARTA [22], the state-of-the-art search technique that has been applied in this domain.

**Blueprint Training.** We train blueprint policy using independent Q-learning with a distributed learning setting. The entire process consists of two modules that run simultaneously. The first is a simulation module that runs $M$ copies of the Hanabi game distributed across $N$ parallel running workers. The simulation module collects AOH $\tau_T$ at the end of each game and writes those data into a centralized prioritized replay buffer [31, 17]. Meanwhile, a training module samples from the replay buffer to train the policy and send a new copy of the policy to the simulation module every $K$ gradient steps. We set $M = 6400$, $N = 80$ and $K = 10$, the first two of which are chosen to make the replay buffer write speed of the simulation module roughly the same as the replay buffer read speed of the training module. We use the canonical techniques such as double Q-learning [46] and dueling DQN [48] for better performance. We use the same Public-LSTM network architecture as in [20] to simplify the belief update procedure. For simplicity, we use independent Q-learning [44]. We train the blueprint policy for 2 million gradient steps/batches and each batch contains 128 AOHs $\tau_i$. Our blueprint gets 24.23 points on average, which is comparable to the strongest blueprint (24.29) in [20] which is trained with a value decomposition network.

**Results.** We present our results in Table 1. For single-agent RL search we set the search horizon $H = 3$, the number of gradient steps $G = 5000$, the number of evaluations for comparing fine-tuned policy against blueprint $E = 10,000$ and the deviation threshold $\epsilon = 0.05$. For multi-agent RL search we set $H = 1$, $G = 10000$, $E = 10,000$, and $\epsilon = 0.035$. For comparison we also run both the single-agent and multi-agent version of SPARTA on our blueprint. Multi-agent SPARTA can be extremely expensive to run depending on the maximum size allowed for range-search, so we pick the hyper-parameter to make it consume roughly the same amount of time per game as multi-agent RL search. Each cell in Table 1 is evaluated on 2000 games. The numbers on top shows the average score and standard error of the mean, and the number at the bottom of each cell is the percentage of winning games (perfect score games). In normal Hanabi, single-agent RL search beats single-agent SPARTA and multi-agent RL search beats multi-agent SPARTA. However, the scores are close enough that the differences are difficult to distinguish. This is likely because the blueprint in normal Hanabi is already extremely strong, so the benefit of additional search is diminished. On the harder 2-hint variant, RL search, and especially multi-agent RL search, significantly outperforms SPARTA. This indicates that searching for joint deviations is helpful, and RL search is a scalable way to find them.

Whether or not RL search is preferable to SPARTA depends in part on the computational budget. Single-agent SPARTA need only search over about 20 possible actions, so it takes 4 seconds to make a move using 5 CPU cores and 1 GPU. For comparison, single-agent RL search would take 69 seconds per move when searching one move ahead with 20 CPU cores and 2 GPUs. However, RL search scales much more effectively. The branching factor in Hanabi is around 500 and the game involves a high amount of stochasticity due to new cards being drawn, so searching even 3 moves ahead is essentially intractable for single-agent SPARTA. In contrast, single-agent RL search that looks 3 moves ahead takes only 88 seconds per move. Similarly, searching two moves ahead (one for each player) in multi-agent SPARTA would mean searching over about $20^{20}$ joint policies, which would be intractable. In contrast, multi-agent RL search is able to do this while taking only 180 seconds per move.

| Variant | Blueprint | SPARTA (Single) | SPARTA (Multi) | RL Search (Single) | RL Search (Multi) |
|---|---|---|---|---|---|
| Normal | $24.23 \pm 0.04$ 63.20% | $24.57 \pm 0.03$ 73.90% | $24.61 \pm 0.02$ 75.46% | $24.59 \pm 0.02$ 75.05% | $\mathbf{24.62 \pm 0.03}$ **75.93%** |
| 2 Hints | $22.99 \pm 0.04$ 17.50% | $23.60 \pm 0.03$ 25.85% | $23.67 \pm 0.03$ 26.87% | $23.61 \pm 0.03$ 27.85% | $\mathbf{23.76 \pm 0.04}$ **31.01%** |

Table 1: **Performance on Hanabi**. Each cell is averaged over 2000 games. The number in the upper half of the cell is the average score $\pm$ standard error of mean (s.e.m.) and the number in the lower half is the percentage of winning games where agents score 25 points.

## 4.2 Atari

We demonstrate the generality of our approach by comparing policy gradient fine-tuning to MCTS in two Atari environments, Ms. Pacman and Space Invaders [6]. Specifically, we aim to answer the following questions:

1. *Does RL Fine-Tuning outperforms MCTS in terms of search time and sample complexity?* Yes, RL Fine-Tuning obtains higher scores in Ms.Pacman than MCTS for a smaller search time budget and a smaller number of samples per step.

2. *Does RL Fine-Tuning performs well even with a weak blueprint?* Yes, RL Fine-Tuning obtains strong results with a weak blueprint in Ms.Pacman and improve the policy in a much more sample-efficient way than carrying the PPO training of the weak policy.

3. *Are the optimal hyperameters robust across different environments?* Yes, an ablation study on the search horizon hyperparameter reveals that the optimal search horizon is the same for Ms. Pacman and Space Invaders.

### 4.2.1 Implementation

**Blueprint Training.** We train a PPO [34] agent until convergence in Ms. Pacman and Space Invaders. In both cases $10^7$ samples are necessary to converge to the optimal PPO blueprint. We also save a weak blueprint after $2.10^6$ samples in Ms.Pacman to answer question 2. The weak blueprint obtains a score that is 5 times smaller than the optimal PPO policy.

**MCTS.** At every testing time step, we build a tree starting from the current state. We use the blueprint policy to guide the action selection during the tree traversal and we use the value network every time we reach a node never seen before or we reach the depth limit of the tree. In our experiments we use a depth limit of 100. We can significantly improve the performance of MCTS by allowing for an additional hyperparameter to balance policy prior and visitation counts in the second term:

$$\arg\max_a Q(s,a) + c \cdot \pi_\theta(a|x)^\beta \cdot \frac{\sqrt{\sum_{a'} N(s,a')}}{1 + N(s,a)} \tag{8}$$

We obtain the best performance with $c = 5$ and $\beta = 0.1$.

**Policy Gradient Fine-Tuning.** Our method achieves a small average time budget by amortizing the search time across multiple steps. In Ms. Pacman, we solve a finite-horizon problem of horizon 30 and we only need to replan every 30 steps (Algorithm 2). We have also tried to amortize MCTS across multiple steps, where we update the tree only after 30 steps. In this setting however, the episode return is worse than what is achieved by the blueprint, emphasizing the need to replan at every timestep, which is not necessary when performing RL search. To optimize an infinite-horizon problem rather than a finite-horizon problem, we can use the refined value instead of the blueprint for the last step of each trajectory. The problem is still simplified due to the biased initial state distribution. In Ms. Pacman, we have observed that this setting leads to similar improvements.

### 4.2.2 Results

**RL Fine-Tuning outperforms MCTS for a fixed search time budget.** Both MCTS and policy gradient fine-tuning are multi-step improvement operators that optimize objective (4) using a value

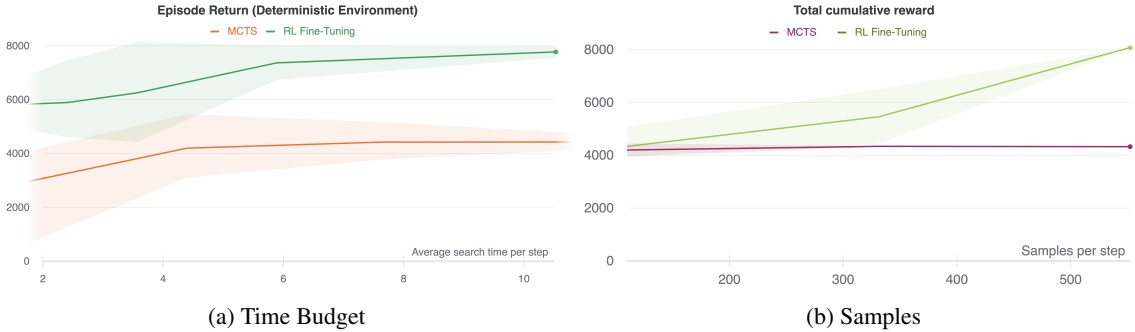

| | (a) Time Budget | | | (b) Samples | |

Figure 1: **MCTS vs RL Fine-Tuning.** (a)When the average time budget is on the order of 1-10 seconds, RL Fine-Tuning consistently outperforms MCTS. (b)RL Fine-Tuning also outperforms MCTS in terms of sample efficiency. The shaded area represent the min/max range across 5 seeds. The curves are smoothed with an exponential moving average.

| Additional Samples | 0 | $3.10^5$ | $4.10^5$ | $8.10^5$ |
|---|---|---|---|---|
| RL Fine-Tuning | 1880 | **3940** | **4580** | **5510** |
| PPO Training | 1880 | 1900 | 1900 | 1920 |

Table 2: **Performance on Ms. Pacman with a weak blueprint.** It is more sample efficient to use RL Fine-Tuning to improve a weak blueprint rather than carrying on the PPO training.

estimate to truncate the objective. Therefore we expect both methods to achieve the same asymptotic performance. We compare both methods with a finite time budget of the order of 10s. Figure 1 shows the return achieved by the agent when performing either MCTS or RL search at action selection time, versus the average search time budget. We see that RL search consistently outperforms MCTS, contrasting with recent work showing that policy gradient was worse than MCTS for planning in the game Hex [3]. The difference of performance with this previous work might be due to the fact that they are using vanilla PG while we are using PPO.

**RL Fine-Tuning is more sample efficient than MCTS.** With RL search, we need an average of 621 samples and 1.2 seconds per step to achieve a return of 8080 which is more than 2 times the return achieved by our asymptotic PPO policy. The total number of additional samples needed is 502,000, which is less than 5% of the samples needed for the blueprint PPO policy to converge. In contrast, MCTS requires an average of 4489 samples per step to reach a cumulative reward of 5820.

**RL Fine-Tuning obtains strong results even with a weak blueprint.** We run experiments in the Ms. Pacman environment using poorer blueprints and compare the average cumulative reward obtained by continuing the PPO training versus performing RL Fine-Tuning for the same number of additional samples. For a blueprint trained during 2000 epochs of 1024 samples (around 1/5 of convergence and obtaining an average cumulative reward of 1880), RL fine-tuning can reach an average cumulative reward of 5510 with an online search using on average 1145 samples per step. In contrast, continuing the PPO training of the blueprint using the same number of additional samples used by RL fine-tuning yields a policy that reach an average cumulative reward of 1920 only (see table 2). We also test a randomly initialized blueprint: RL fine-tuning can reach an average cumulative reward of 2730 with an online search using on average 1360 samples per step while the PPO training of this blueprint with the same number of additional samples yields an average cumulative reward of 1280 only (see table 3).

**The hyperparamters of RL Fine-Tuning are robust across different environments.** After performing the same ablation study on the horizon in both Ms. Pacman and Space Invaders, we have found that the optimal horizon is 32 for both environments. Thus there is reason to think that this value is nearly optimal in several other Atari games and readers willing to apply our method to other Atari games should start experimenting with this value.

| Additional Samples | 0 | $2.10^5$ | $4.10^5$ | $8.10^5$ |
|---|---|---|---|---|
| RL Fine-Tuning | 60 | **1180** | **1800** | **2730** |
| PPO Training | 60 | 689 | 732 | 1280 |

Table 3: **Performance on Ms. Pacman with a random blueprint.** RL Fine-Tuning also outperforms PPO in term of sample efficiency when the blueprint is randomly initialized.

## 5 Related Work

**Fine-tuning in supervised learning.**  Instead of searching for the optimal action sequence in the action space, which is what MCTS does, we are effectively searching in the parameter space of the neural network. We do that by following the gradient of the truncated objective. Since the truncated objective and the biased data distribution are close to the objective and the data distribution the blueprint neural network was trained on, the performance of our method should reflect the success of neural network fine-tuning in supervised learning [43, 30].

**Fine-tuning in reinforcement learning.**  Replacing action search with neural network parameter search is reminiscent of recent work in continuous control performing cross-entropy method in the parameter space of the neural network [47]. They argue that the improvement over searching in the action space is due to the smoothness of the neural network loss landscape [24]. We take this idea further by making full use of the parameter space. While they are using scalar information – the return of each rollout –, we use richer information – the gradient of the return with respect to $\theta$. By doing so, we argue that we lose less information about the blueprint than other online planning methods, which enables us to stay efficient in high dimensions.

Policy gradient fine-tuning is related to recent work on iterative amortized policy optimization [25]. Considering policy optimization in the lens of variational inference [23], they argue that amortizing the policy into a neural network leads to the same suboptimal gap than amortizing the posterior in VAEs and propose to inject gradient information into the policy network's input at action selection time to close the gap. While their policy network meta-learns how to use the gradient, we use a fixed update rule. Meta-learning the update rule is a promising future direction that could be stacked on top of our method. However, we suspect learning the update rule at blueprint training time to be intractable in complex environments such as Hanabi.

A large range of previous work has proposed to combine RL with tabular search [40]. In Expert Iteration [2, 32], a policy network is used to amortize an expert tree search algorithm using imitation learning. Rather than amortizing the traversal policy, Hamrick et al. propose to train a Q-network via model-free RL to amortize the Q-values returned by MCTS [16]. Our method differs from this line of work because we don't use any tree search for maximum scalability.

Finally, there has been previous work investigating some variants of RL search. Temporal-difference search was introduced in [35] and applied to Computer Go. Policy gradient search [3] performs vanilla policy gradient to fine-tune a rollout policy and selects the action by performing MC rollouts with this policy. They perform worse than MCTS in the game Hex. Springenberg et al. [40] compare a variant of policy gradient fine-tuning to tree search in continuous control environments and show that PG fine-tuning performs almost as well as tree search, but not equally well. By contrast to these rather negative results, our positive results show that RL search outperforms MCTS in Ms. Pacman and achieves state-of-the-art performance in Hanabi. Additionally, prior work has looked to generalize between states during MCTS, though without fine-tuning of a pre-trained network [49].

## 6 Discussion

Despite the incredible success of neural network function approximation in reinforcement learning environments [26, 15, 4], search algorithms have largely remained tabular, as can be seen for example in AI algorithms for go [36, 37, 38], poker [27, 7, 8, 9], and Atari [32, 33, 21]. Tremendous research has gone into adding heuristics that enables tabular search to perform better in these domains, especially in perfect-information deterministic environments [14, 11]. However, we argue

that in order to scale to larger, more realistic environments, especially those involving stochasticity and partial observability, it is necessary to move past purely tabular search algorithms.

The jump from tabular search to RL search comes at a steep upfront cost. However, we show in this paper that in sufficiently complex environments, including the established AI benchmark Hanabi, RL search enables state-of-the-art performance and scales much more effectively. We show this also holds in the Atari Ms. Pacman environment, and we expect it to be true in a wide variety of other domains as well. As computational costs continue to decrease, it is likely that the relative performance of RL search will improve even further. To quote Richard Sutton's *The Bitter Lesson* [41], "One thing that should be learned from the bitter lesson is the great power of general purpose methods, of methods that continue to scale with increased computation even as the available computation becomes very great. The two methods that seem to scale arbitrarily in this way are search and learning."

While a lot of methods combining RL with tabular search have been developed, we have shown that we can achieve better results by completely amortizing online planning into a neural network via RL fine-tuning. Most importantly, we have shown that RL search can effectively search complex domains where tabular search is intractable. This enables us to achieve state-of-the-art results in the AI benchmark Hanabi, which features both stochasticity and partial observability, both of which are prevalent in the real world.

**Limitations:** While we achieve good results in discrete environments, our algorithms are currently focused on discrete environments. A promising direction for future work would be an investigation of similar techniques in continuous control domains and a comparison to existing techniques (mentioned in the main text) in those domains. A potential limitation of our method compared to gradient-free methods is that we might be constrained by the local minimum of a too weak blueprint. In other words, we may be unable to discover drastically new solutions to the control problem that an optimal global method solving the problem from scratch would find.

**Broader impact:** RL search is a decision-time planning method aimed at improving search in very high-dimensional spaces. There has been a rich history of previous work in this area, where the goal has been to develop more efficient search methods. This work is in line with previous work and should not directly cause broader harm to society. We are not using any dataset or tool that will perpetuate or enforce bias. Nor does our method make judgements on our behalf that could be misaligned with our values.

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
