# OpenReview forum: "Scalable Online Planning via Reinforcement Learning Fine-Tuning"
_NeurIPS.cc/2021/Conference — NeurIPS 2021 Poster_

### Official Review · Reviewer_Ypiw · 2021-07-15

**Rating:** 6
**Confidence:** 3

**Summary:**

This paper proposes to replace tabular search methods like MCTS with fine tuning of neural policies for online planning.

The motivation is that fine tuning the neural policies online may be more efficient than doing MCTS which has no generalization between states during the search.

The key algorithmic contribution is the biased RL objective used for more efficient online fine tuning of the neural policy.

**Limitations And Societal Impact:**

My main concern is that this paper does not have enough algorithmic contributions.

My second concern is that this method may only work well when the given blueprint policy is well trained. This is because the key potential advantage of this method over tabular search methods is the effective generalization between states and actions. This is not possible with a poorly trained neural network. In the Atari experiments, a PPO agent is trained to convergence. I would be very interested to see how would this method perform when given blueprint policies that are not so well trained. I am also wondering why there is still a performance gain from online fine tuning given that the PPO agent has already been trained to convergence? Does this imply that the training of the PPO agent is not optimal or perhaps a larger network can be used? Nevertheless, I think requiring a good blueprint policy to be fine tuned over should be a major limitation of this method.

My third concern is that the method has only been tested and compared on Hanabi and one Atari game. More experimental results from other domains, for example other Atari games, will make the results more convincing.

**Main Review:**

Originality: I find the idea of replacing MCTS with online fine tuning of neural policies interesting. However, I don't think overall the paper has enough algorithmic contributions.

Quality: The method seems correct. Limitations see below.

Clarity: The paper is easy to follow.

Significance: As mentioned, I think this is an interesting direction. Also, I think the general problem setup of this paper, which is how to make good online decisions with a blueprint policy and a model important to study.

**Time Spent Reviewing:**

6

---

> ### Author Response · Authors · 2021-08-10
> **Review Response**
>
> Thanks for taking time to review our paper!
> ***
> >*My main concern is that this paper does not have enough algorithmic contributions.*
>
> The main contribution of this paper is that by using RL fine-tuning, we are able to conduct search/planning in situations that are intractable for traditional tabular search methods (such as MCTS and SPARTA). The paper shows that neural network based search/planning method with RL is more scalable and performs well. We believe this is an important contribution and can potentially attract new attention for this direction of research. The algorithm itself is conceptually uncomplicated, which we believe is a strength of the paper.
> ***
> >*My second concern is that this method may only work well when the given blueprint policy is well trained. This is because the key potential advantage of this method over tabular search methods is the effective generalization between states and actions.*
>
> The quality of the blueprint affects the performance of both tabular search and RL fine-tuning search, because both tabular search and RL fine-tuning search require a value function for leaf node evaluation. This can either come from the blueprint value network, or from rolling out the blueprint policy network to the end of the game. While it is the case that RL fine-tuning search fine-tunes the blueprint policy, its performance is not bounded by the quality of the blueprint. For example, in principle we could start the policy network in every subgame as a brand new uninitialized network and learn from scratch.
>
> Furthermore, the main focus of this paper is to enable search in situations where existing tabular search methods fail to scale and to achieve new state-of-the-art results that RL alone is not able to realistically achieve. Therefore, we believe the results with a strong blueprint are the most interesting results.
> ***
> >*I would be very interested to see how would this method perform when given blueprint policies that are not so well trained.*
>
> We ran additional tests with a very poor blueprint (an early snapshot with only 1 hour of training) that gets 14.4 in Hanabi. SPARTA (using rollouts to the end of the game) improves it to 18.05 while single-agent RL fine-tuning search (using the learned Q value network) achieves a comparable result of 17.71. This is a positive result for RL fine-tuning search because SPARTA uses rollouts to the end of the game (i.e., a perfect value network without any function approximation error) while in this experiment RL fine-tuning search only uses the Q value estimates from the poor-quality blueprint. Rollouts result in better performance, but are much more expensive especially for long games. It is possible to further improve the performance of RL fine-tuning by either using rollouts to the end of the game (like in SPARTA) or by training a more accurate value function with supervised learning.
>
> We also ran additional tests in the Ms. Pacman environment using poorer blueprints and compared the average cumulative reward obtained by continuing the PPO training versus performing RL fine-tuning search for the same number of additional samples. For a blueprint trained during 2000 epochs of 1024 samples (around 1/5 of convergence and obtaining an average cumulative reward of 1800), RL fine-tuning can reach an average cumulative reward of 5510 with an online search using on average 1145 samples per step. In contrast, continuing the PPO training of the blueprint using the same number of additional samples used by RL fine-tuning yields a policy that reach an average cumulative reward of 1920 only. We also tested a randomly initialized blueprint: RL fine-tuning can reach an average cumulative reward of 2730 with an online search using on average 1360 samples per step while the PPO training of this blueprint with the same number of additional samples yields an average cumulative reward of 1280 only.
> ***
> >*I am also wondering why there is still a performance gain from online fine tuning given that the PPO agent has already been trained to convergence? Does this imply that the training of the PPO agent is not optimal or perhaps a larger network can be used?*
>
> Our method can still increase the performance given a fully converged PPO/DQN network because even though the agent might be fully converged, it is still not optimal (unless there is infinite network capacity and infinite data). Our RL fine-tuning search algorithm effectively augments the network capacity and training data by allowing the network to focus on optimizing performance on just one small part of the game (the subgame) at a time. It trains the policy to act optimally given the particular history encountered during one specific game, so it is potentially a much simpler game with a smaller state space to consider.
> ***
> >*My third concern is that the method has only been tested and compared on Hanabi and one Atari game. More experimental results from other domains, for example other Atari games, will make the results more convincing.*
>
> We made a special effort in our experiments to test on two very different domains: Ms. Pacman (single-agent, fully observable, arcade game) and Hanabi (multi-agent, partially observable, card game). Moreover, within Hanabi we conduct experiments on both single-agent and multi-agent search. At the reviewer's request, we have ran additional experiments in Space Invaders and we obtain similarly convincing results that will be added to the paper. While more experiments are always desirable, it is expensive to conduct thorough, statistically significant experiments using RL fine-tuning because there is a substantial cost at test time in addition to just training time.

---

> ### Author Response · Authors · 2021-08-17
> **Has our response addressed your concerns?**
>
> Hello reviewer Ypiw, we would be grateful if you can confirm whether our response has addressed your concerns, and let us know if any issues remain. To recap our response, we:
> - emphasized that the simplicity of the algorithm combined with the strong results it obtains should attract new attention for this direction of research
> - added additional experiments using poorly trained blueprint that showed that our method does not require strong blueprints
> - performed experiments in a new Atari environment (Space Invaders)  (see comment “New experimental results”)
>
> We also ran additional experiments that show that:
> - RL Fine-Tuning outperforms MCTS in terms of sample efficiency
> - even with a poorly trained blueprint, RL Fine-Tuning obtains strong results
> - RL Fine-Tuning works well in other environments
> - the chosen horizon is also optimal in other environments
>
> Please refer to our comment “New experimental results” posted today for more details on these additional experiments.

---

> > ### Comment · Reviewer_Ypiw · 2021-08-22
> > **Response**
> >
> > Dear authors,
> >
> > Thanks for the detailed rebuttal and the new experimental results. I've changed my mind on the significiance of this paper's contribution and therefore increased my score. Indeed, this paper shows an exciting new direction for online planning with RL instead of MCTS.
> >
> > Maybe I've missed it, but I think one thing that is still missing is: under what circumstances, we can expect fine tuning with RL outperforms MCTS during online planning? The experimental results you have currently clearly demonstrates the cases where RL fine tuning is more effective. However, I think performing a few experiments to test this boundary and show limitations of your method will make the paper stronger. My hypothesis would be that when your blueprint is not so great and there isn't a lot of budget for MCTS planning or RL fine tuning, MCTS may turn out to be more efficient? Curious to get a better understanding of it if possible.

---

> > > ### Author Response · Authors · 2021-08-30
> > > **Additional Experiments: Limitations of RL Fine-Tuning**
> > >
> > > Thank you for suggesting to investigate the limitations of RL Fine-Tuning which should indeed make the paper stronger. We ran additional experiments with a weak blueprint and a small number of samples (range: 1 to 300 samples per step on average) in Ms. Pacman and Space Invaders. Results show that RL Fine-Tuning performs slightly better than MCTS in Ms. Pacman and is on a par with MCTS in Space Invaders. Globally, the superiority of RL Fine-Tuning is indeed less apparent for a small number of samples per step. Furthermore, we hypothesized that MCTS would be more competitive in a fully deterministic game. We ran additional experiments with a small number of samples in Breakout and on this range of samples (1-300), MCTS performs indeed slightly better than RL Fine-Tuning. We will further investigate this direction and add the results to the final version.

---

### Official Review · Reviewer_2XCZ · 2021-07-15

**Rating:** 7
**Confidence:** 3

**Summary:**

This paper proposes to replace a Monte-Carlo tree search style search method by instead fine-tuning a neural network by running reinforcement learning on trajectories generated by a perfect model of the environment starting from the current state at inference time. They apply this idea in a cooperative multi-agent planning in the game Hanabi, as well as in single-agent planning in the game Ms. Pacman where the agent has is given access to the simulator. Experimental results suggest that the fine-tuning procedure is able to perform similarly to or better than a strong tabular search method (SPARTA) in Hanabi. In Atari, they demonstrate that the procedure seems to outperform MCTS, particularly as the total search budget is made larger for each algorithm.

**Limitations And Societal Impact:**

Limitations are reasonably well covered. I don't see much potential for negative societal impact.

**Main Review:**

I feel the approach considered in this work is quite interesting and important. It makes sense that ultimately we will want to replace MCTS like procedures with something that generalizes more naturally during the search procedure itself. The experiments presented here seem like a good proof of concept for such methods and the fact that they include results for both a single agent game and a multi agent benchmark is a plus.

My main uncertainty is that the experimental details included are a bit lacking (though at least the code is made available). In particular, see 'Questions about Experimental Details' heading below.

Also, the paper could be made more self-contained in terms of ensuring that all the variables that are used are clearly defined (see Minor Comments and suggestions).

I also am not very knowledgeable of the relevant multi-agent search literature to evaluate the multi-agent portion of the work. But from what I can tell it seems generally well done.

Questions about Experimental Details
====================================

Some details of the experimental methodology such as various hyper-parameters and how they were selected seem to be missing:

Line 131: 'the transitions used to fine-tune the Q-network are sampled with probability p from the global buffer replay' what is this p and how is it chosen? Also, why is the global replay buffer and how is it generated?

Line 236: 'We obtain the best performance with c = 5 and $\beta$ = 0.1' what other values were searched over here?

What hyper-parameters were used for the blueprint Q-learning and PPO agents in Hanabi and Ms. Pacman respectively and how were they chosen?

Minor Comments and Suggestions
==============================

Line 33: 'non-tabular planning algorithms improve performance'->'non-tabular planning algorithms to improve performance'

Between Line 59-60 (line numbers missing here for some reason): is $\Omega^i(\tau_0)$ defined somewhere?

Line 137: Is $B(\tau_t^i)$ explicitly defined somewhere? From looking at the cited SPARTA paper it seems like this should maybe be $B^i(\tau_t)$ if the notation is supposed to be consistent with that work.

Algorithm 1: I think t is kind of overloaded here in that $s_t$ is the state at the current time but $\delta_t$ is $\forall t\in[0,H-2]$ using different variable names would make this clearer.

Line 260: 'like MCTS do'->'like MCTS does'

For the Ms. Pacman results, it would be interesting to see how MCTS and RL-fine-Tuning compare in terms of total number of states expanded in addition to average search time.

Update
======
I thank the reviewers for their clarifications. My empirical concerns are addressed and I will maintain my score. The additional experiments are also welcome.

**Time Spent Reviewing:**

8

---

> ### Author Response · Authors · 2021-08-10
> **Review Response**
>
> Thanks for your valuable feedback!
> ***
> Re “line 131”: In the experiments we ended up creating a new replay buffer each time we fine-tune, so p is effectively zero. The purpose of introducing a non-zero probability of sampling the buffer used during the training of the blueprint (what we call the global buffer) was to stabilize the fine-tuning but we have observed that it is not necessary in practice.
> ***
> Re “line 236”: We performed a grid search with 5 values of c (0.1, 0.3, 1, 5, 10) and 5 values of beta (0.1, 0.5, 1, 2, 10).
> ***
> Re “hyper-parameters”: For the training blueprint in Hanabi, we follow the exact hyper-parameters of previous work [1] except that we use independent Q learning and they use value decomposition. For the PPO agent, we use the Atari hyperparameters used by the OpenAI baseline.
> ***
> Re "line 59-60": The $\Omega_i$ is the observation function defined in Section 2.1.
> ***
> Re “line 137”: $B(\tau_t^i)$ here is the $B^i(\tau_t)$ in the SPARTA paper [2]. We apologize for the missing definition and we'll definitely add them in the next version.
> ***
> Re “Algorithm 1”:  t was indeed overloaded, thanks for pointing that out. We will change to $\delta_{t’}$.
> ***
> Re "For the Ms. Pacman results, it would be interesting to see how MCTS and RL-fine-Tuning compare in terms of total number of states expanded in addition to average search time." : We have added additional results to the paper that compare MCTS and RL Fine-Tuning in term of number of states expanded. The additional results show that RL fine-tuning also outperforms MCTS in terms of sample efficiency. For example, it takes MCTS an average of 4489 states expanded per step to reach a cumulative reward of 5820 while it takes RL Fine-Tuning only an average of 621 samples per step to reach a cumulative reward of 8080.
> ***
>
> [1] H. Hu, A. Lerer, N. Brown, and J. N. Foerster. Learned Belief Search: Efficiently Improving Policies in Partially Observable Settings, 2021. URL https://arxiv.org/abs/2106.09086
>
> [2] A. Lerer, H. Hu, J. N. Foerster, and N. Brown. Improving Policies via Search in Cooperative Partially Observable Games, 2020. URL https://arxiv.org/abs/1912.02318

---

### Official Review · Reviewer_VMv6 · 2021-07-16

**Rating:** 7
**Confidence:** 4

**Summary:**

This paper proposes a new planning method, aiming at improving performance in a large and realistic environment. The proposed method performs as a planning module and can be integrated with different types of online learning agents. Different from tabular search algorithms, the method formulates the planning step as a reinforcement learning problem and updates the policy or action-value function by bootstrapping. Thus, the method reduces the horizon of search and gains learning efficiency in large search spaces and continuous control problems.

**Limitations And Societal Impact:**

Yes.

**Main Review:**

The proposed method improves search efficiency and is applicable in large search spaces. However, the discussion on how key parameters affect the performance is missing. It is also not clear if the hyperparameter setting in the submitted codebase is the same as the setting used for generating the result.

More details are provided below.

The paper discusses different cases separately in detail, including policy gradient methods and action value methods, as well as single-agent and multi-agent settings. This suggests the method can be integrated with different types of reinforcement learning agents.

The reported performance in Table 1 is averaged over 2000 games, which gives a reliable averaged value. The experiment reported in Figure 1 tests 5 seeds. Testing more seeds would be better, though 5 is acceptable.

The idea of truncating the target is similar to n-step bootstrapping (Chapter7 in [1]). I understand this paper focuses on planning instead of value function learning, though adding references about n-step bootstrapping might help the reader to understand the main idea.

The trajectory length H seems like a key parameter in the algorithm. If H equals the episode length, the update would be the same as MCTS, which means the search goes until the end of the episode. I assume this parameter needs to be tuned carefully in the experiment. It would be nice if there is an experiment discussing how performance changes regarding different search horizons. If the length of the horizon can be fixed and no need to tune, it will also be good to explain how this number should be chosen. This will make it easier for readers to apply this method to other problems.

It may be worth listing hyperparameter settings in the paper, such as network structure and learning rate for both the proposed method and baselines, for reproducibility. I understand that the code base is provided and there are default settings in the code, but it is not clear to me if the default setting is the same setting used for generating the results reported in the paper.

Small Things:
In Algorithm 1, the symbol used in output ($\tilde{\theta}$) is inconsistent with the return ($\theta_N$)

[1] Sutton, Richard S., and Andrew G. Barto. Reinforcement learning: An introduction. MIT press, 2018.


**Time Spent Reviewing:**

5

---

> ### Author Response · Authors · 2021-08-10
> **Review Response**
>
> Thank you for your review!
> ***
> >*The proposed method improves search efficiency and is applicable in large search spaces. However, the discussion on how key parameters affect the performance is missing. It is also not clear if the hyperparameter setting in the submitted codebase is the same as the setting used for generating the result.*
>
> For the blueprint training and RL fine-tuning, we largely followed previous open sourced work for the hyper-parameters except for the frequency of target network synchronization because we want to update the target more frequently given that the network is only trained for 5K batches. Important hyper-parameters related to search such as horizon H and deviation threshold are mentioned in the paper. We will add a comprehensive list of the hyperparameters in the final version.
> ***
> >*The idea of truncating the target is similar to n-step bootstrapping (Chapter7 in [1]). I understand this paper focuses on planning instead of value function learning, though adding references about n-step bootstrapping might help the reader to understand the main idea.*
>
> The main motivation here for truncating the target is to make training computationally feasible for complex environments like Hanabi. The RL search algorithm in principle does not depend on the fact that there is such a pre-existing Q function to bootstrap. Thank you for pointing out the connection to n-step bootstrapping and it would indeed help readers to understand the idea. We will add references to this in the final version of the paper.
> ***
> >*The trajectory length H seems like a key parameter in the algorithm. If H equals the episode length, the update would be the same as MCTS, which means the search goes until the end of the episode. I assume this parameter needs to be tuned carefully in the experiment. It would be nice if there is an experiment discussing how performance changes regarding different search horizons. If the length of the horizon can be fixed and no need to tune, it will also be good to explain how this number should be chosen. This will make it easier for readers to apply this method to other problems.*
>
> Search horizon H controls the depth of the search. It decides the length of the trajectory on which we fine-tune the policy at test time as well as the number of turns in which the tuned policy will be used if it exceeds the deviation threshold. Therefore, setting H to the length of the episode does not resemble MCTS at all. Instead, it would basically be the same as training a RL policy to play until the end of the game given the current game state (or, in an imperfect-information game, the current public belief state). The best value for H depends primarily on the available computational budget. Deeper search requires more samples and computation in the fine-tuning/search process. Whether or not this increased computational cost is worthwhile depends on the nature of the game. In Hanabi, we found that it did not meaningfully change the experimental results, likely because Hanabi is a very “local” game that does not require much planning beyond the next few turns. Sometimes we may also limit the network capacity available for tune-tuning, e.g. only tuning the last layer. In those cases the H should not be set to a large value since the network does not have enough capacity to capture longer term planning. Thanks for raising this question and we will add more discussions on this topic to the paper and add our results showing that, at least in Hanabi, searching deeper does not meaningfully change the results.
>
> We also ran additional experiments for Ms. Pacman comparing the cumulative reward obtained for a wide range of search horizons (from 1 to 512, the best performing search horizon is 32) and will add the results to the paper.
> ***
> >*It may be worth listing hyperparameter settings in the paper, such as network structure and learning rate for both the proposed method and baselines, for reproducibility.*
>
> We will add a comprehensive list of the hyperparameters in the final version.

---

> ### Author Response · Authors · 2021-08-17
> **Has our response addressed your concerns?**
>
> Hello reviewer VMv6, we would be grateful if you can confirm whether our response has addressed your concerns, and let us know if any issues remain. To recap our response, we:
> - added a comprehensive list of the hyperparameters
> - added references to n-step bootstrapping
> - added an ablation study of the horizon in both Ms. Pacman and Hanabi
> - added an ablation study of the horizon in Space Invaders (see comment “New experimental results”)
>
> We also ran additional experiments that show that:
> - RL Fine-Tuning outperforms MCTS in terms of sample efficiency
> - even with a poorly trained blueprint, RL Fine-Tuning obtains strong results
> - RL Fine-Tuning works well in other environments
> - the chosen horizon is also optimal in other environments
>
> Please refer to our comment “New experimental results” posted today for more details on these additional experiments.

---

> > ### Comment · Reviewer_VMv6 · 2021-08-17
> > **Discussions**
> >
> > Hello! Yes, my concerns are addressed. Thank you for the response and new experiments. I would like to increase my score.

---

### Author Response · Authors · 2021-08-16
**New experimental results**

We have conducted a series of new experiments based on feedback from reviewers which will be added to the final version of the paper. We summarize them below:
- **RL Fine-Tuning outperforms MCTS in terms of sample efficiency**: Reviewer 2XCZ asked for an additional experiment comparing RL Fine-Tuning and MCTS in terms of sample efficiency. It was a very good idea since the results of this experiment clearly show the superiority of RL Fine-Tuning over MCTS and will hopefully convince reviewers VMv6 and Ypiw. Indeed, MCTS requires an average of 4489 samples per step to reach a cumulative reward of 5820 while RL Fine-Tuning only requires an average of 621 samples per step to reach a cumulative reward of 8080.
- **Even with a poorly trained blueprint, RL Fine-Tuning obtains strong results**: Reviewer Ypiw asked for an additional experiment using a poorly trained blueprint. We performed additional experiments in Hanabi and Ms. Pacman using poor blueprints and obtained positive results that will hopefully convince reviewer Ypiw that our method can also be used with poorly trained blueprints. For example, for a poorly trained blueprint in Ms. Pacman, RL Fine-Tuning obtains nearly 3 times the reward that we get if we continue the PPO training of the blueprint with the same number of additional samples (5510 vs.1920). See our response to reviewer Ypiw for additional results with poorly trained blueprints.
- **RL Fine-Tuning works well in other environments**: Reviewer Ypiw asked for an additional experiment on another Atari environment. We have performed new experiments in Space Invaders and the positive results will hopefully convince reviewer Ypiw that our method can be applied to a wide range of environments. Indeed, RL Fine-Tuning obtains a cumulative reward of 1685 in Space Invaders with a poorly trained blueprint (achieving a cumulative reward of 280) and using only an average of 242 samples per step.
- **The chosen horizon is also optimal in other environments**: Reviewer VMz6 asked how the horizon should be chosen by readers willing to apply our method to other environments. We have used the newly introduced Space Invaders environment to bring a first answer to the question. Indeed, after performing the same ablation study on the horizon as in Ms. Pacman, we have found that the optimal horizon is 32 for both environments. Thus there is reason to think that this value is nearly optimal in several other Atari games and readers willing to apply our method to other Atari games should start experimenting with this value.

---

### Decision · Program_Chairs · 2021-09-27

**Decision:**

Accept (Poster)

**Comment:**

The paper proposes to replace tabular MCTS-like search methods with finetuning a pretrained policy for the states encountered online. The the high-level method is a good contribution despite being straightforward. The experiments are fairly limited though, for what is largely an empirical paper. Even with the addition of a few more Atari games as a result of the post-rebuttal discussion, the evaluation is weak by the standards of empirical RL papers. I'd expect at least 20-30 more Atari games, or the Atari games on which the authors already have the results and the Procgen suite on top of that. I'm not discounting Hanabi -- it's a hard benchmark, but it's still just one benchmark. Running experiments on more games/environments is especially important since, according to the additional results mentioned in the discussion, in low-data and weak-blueprint regimes RL-Fine-Tune's advantage over MCTS is far from apparent.

On balance, I cautiously recommend acceptance since RL-Fine-Tune seems novel and a strong baseline, and due to its simplicity it's likely to be experimented with and built on by others.